# Prevalence and characteristics of cannabis-induced toxicoses in pets: Results from a survey of veterinarians in North America

**Richard Quansah Amissah**[1], **Nadine A. Vogt**[2], **Chuyun Chen**[1], **Karolina Urban**[3], **Jibran Khokhar**[1]*

**1** Department of Biomedical Sciences, Ontario Veterinary College, University of Guelph, Guelph, Ontario, Canada, **2** Department of Population Medicine, Ontario Veterinary College, University of Guelph, Guelph, Ontario, Canada, **3** Avicanna Inc., Toronto, Canada

* jkhokhar@uoguelph.ca

**Data Availability Statement:** All dataset and python codes are available from the Open Source Framework repository (accession number: DOI 10.17605/OSF.IO/SCRMY).

## Abstract

Cannabis legalization in North America has coincided with an increase in reports of cannabis-induced toxicosis in pets, but the magnitude of this problem, as well as outcomes of these incidents remain unknown. Therefore, we examined the frequency, diagnostic criteria, clinical signs, and prognoses of cannabis toxicoses in pets in North America. We conducted an online survey between January, 2021 and April, 2021 targeting veterinarians practicing in Canada and the United States (US). Out of the 251 study participants, 191 practiced in Canada. Cannabis toxicosis was most commonly reported in dogs (n = 226 veterinarians), and the number of toxicosis cases increased significantly in Canada (p<0.0001) and the US (p = 0.002) after October, 2018. Frequently reported clinical signs of cannabis toxicosis included: urinary incontinence (n = 195), disorientation (n = 182), ataxia (n = 178), lethargy (n = 150), hyperesthesia (n = 134), and bradycardia (n = 112). Edibles were most commonly suspected to be the cause of toxicosis (n = 116). The most common route of exposure was ingestion (n = 135), while the most cited reason was ingestion while unattended (n = 135). Cannabis toxicosis was mostly diagnosed using supportive clinical signs (n = 229), the most common treatment was outpatient monitoring (n = 182), and pets were most often treated as out-patients (n = 103). The legalization of cannabis use in Canada and the US is likely an important factor associated with the increased cannabis toxicosis cases in pets; however, the legal status may also increase reporting. The medicinal use of cannabis by pet-owners for pets may also contribute to a portion of the reported toxicoses. Most pets that experienced cannabis toxicosis recovered completely, suggesting that most cannabis toxicoses do not result in long-term ill effects. Even though some deaths (n = 16) were reported in association with cannabis toxicosis, the presence of confounders such as toxins, and underlying conditions cannot be ruled out, emphasizing the need for rigorous controlled laboratory studies to investigate this important issue.

**Funding:** This research was funded by a Natural Sciences and Engineering Research Council Alliance Grant (ALLRP 549529 to JYK; https://www.nserccrsng.gc.ca/innovate-innover/alliance-alliance/index_eng.asp) and a MITACS Accelerate Fellowship (IT27597 to RQA and JYK; https://www.mitacs.ca/en/programs/accelerate/fellowship) in partnership with Avicanna Inc. The funders had no role in the design, data collection and analysis, decision to publish, or preparation of the manuscript.

**Competing interests:** Dr. Urban is an employee of Avicanna Inc., during which time she has received stock options. Avicanna Inc. did not influence the design, conduct or interpretation of the data derived from this study. This does not alter our adherence to PLoS One policies on sharing data and materials. None of the other authors have competing interests.

## Introduction

With the widespread legislative changes legalizing cannabis across most of North America, cannabis has become the object of considerable public health and policy discussions [1]. The increased accessibility to cannabis has prompted an increased interest for its therapeutic value in human and, more recently, veterinary medicine [2]. In fact, the sales of cannabis products for pets have increased by 1000% between 2016 and 2017 and a survey found that 79.8% of Canadians have previously bought cannabis products for their dog(s) [3]. Although research is ongoing, there are only a handful of published studies that examine the clinical use of cannabis in veterinary medicine, and even fewer have examined basic pharmacokinetic and toxicology data [4]. Due to this, the education of veterinarians and pet owners is hindered, resulting in intentional or accidental cannabis exposure of pets without proper oversight or knowledge. In fact, a study in Colorado found a strong correlation between the number of registered medical cannabis cardholders and cases of cannabis toxicosis in dogs, with a 4-fold increase in reported cases between 2005 and 2010 [5]. Additionally, over the past 6 years, there was a 448% increase in reports of cannabis poisoning cases in companion animals in the United States (USA) and Canada [6]. The Animal Poison Control Center has also reported a 765% increase in calls regarding pets ingesting cannabis in 2019 compared to the previous year [7]. In Canada, as expected, the total number of cases reported are fewer than in the USA, but they have been increasing since 2018 according to the Canadian Veterinary Medical Association [8].

Taken together, these data suggest that cases of cannabis toxicosis in companion animals are on the rise, warranting further investigations into these incidents. The aim of our study was to gather relevant information from veterinarians in clinical practice regarding cannabis toxicoses. More specifically, our objectives were: a) to examine veterinarian-reported trends in the frequency of cannabis toxicoses pre- and post-legalization October 2018 (date of legalization in Canada); b) characterize diagnostic criteria used for cannabis toxicoses; c) characterize the clinical signs and prognoses of veterinary-reported cannabis toxicoses; d) identify any evidence supporting the lethality of cannabis in companion animals.

## Materials and methods

The need for approval by the Institutional Ethical Review Board was waived due to the nature of the survey questions being only about their veterinary practice, and no personal data were collected. Additionally, the data collected were anonymized and participants were informed on the first page of the survey questionnaire that completing the survey implied consent to participate. To assess cannabis toxicosis in companion animals in both Canada and USA, we designed an online survey using Qualtrics (Provo, Utah, USA). In this study, cannabis refers to substances derived from either the cannabis plant or synthetic cannabinoids. Participants were practicing veterinarians in either Canada or the USA, who were treating pets, and had been presented with cases of cannabis toxicosis. The duration of the survey was from 28th January, 2021 to 30th April, 2021. Before launching the survey, it was pre-tested by the authors and their colleagues, and feedback on the content of the survey was obtained and incorporated into the final version. The Canadian Association of Veterinary Cannabinoid Medicine, Canadian Veterinary Medical Association, Alberta Veterinary Medical Association, Nova Scotia Veterinary Medical Association, Newfoundland and Labrador Veterinary Medical Association, and the Ontario Veterinary College supported with recruitment of participants by distributing the survey to their members. Participants were also encouraged to distribute the survey to their colleagues. The link to the survey was distributed via websites, regular e-newsletters, and magazines of the aforementioned associations. All the data were collected anonymously in Qualtrics. The need for approval by the Institutional Ethical Review Board was waived due to

the nature of the survey questions being only about their veterinary practice, and no personal data were collected. Additionally, the data collected were anonymized and participants were informed on the first page of the survey questionnaire that completing the survey implied consent to participate. Except for questions related to consent to participate and eligibility to participate in the survey, participants could opt not to answer any question, and could decide whether to complete the survey or not. Details of the online survey and the minimal data set can be found in S1 File and S1 Table, respectively. We initially intended to compare the trends in toxicosis cases in Canada and the US; however, the number of US participants was too low for such a comparison to be made. Therefore, while we have described data obtained from veterinarians practicing in the USA in some instances, we have chosen to describe combined data from both Canada and the USA in cases where no differences were observed between the two countries.

## Statistical analysis

At the end of the survey, the data were exported from Qualtrics to Microsoft Excel for analysis. All data were analyzed using libraries in Python (version 3.0). Since the data obtained were categorical, and therefore not normally distributed, non-parametric tests were used for analysis. Unless otherwise stated, the results reported represent the number of participants that responded to a specific question. Descriptive statistics were performed in Python. Comparison of the number of cannabis toxicosis cases pre- and post-legalization was performed using the Wilcoxon-signed rank test in Python. The number of pets treated according to hospital setting and practice type was analyzed using the Kruskal Wallis test followed by the Dunn's post-hoc test in Python. The Chi-squared Goodness of fit test was used to compare the frequencies of each clinical sign, and for clinical signs with significantly different frequencies, a post-hoc binomial pairwise test was conducted. Differences were considered statistically significant when the p-value was less than 0.05. Graphs were plotted using GraphPad Prism version 6.01 (GraphPad Software Inc., La Jolla CA, USA).

## Results

### Demographics

Out of the 251 participants who began the survey, a total of 222 participants completed it (Fig 1A); 7 participants were excluded, as they did not meet the eligibility criteria. Most of the veterinarians practiced in Canada (n = 191; Fig 1B) with the majority practicing in the province of Ontario (n = 108; Fig 1C). Most veterinarians worked in urban areas (Fig 1D) and practiced general medicine (Fig 1E).

Veterinarians reported that cannabis toxicoses were most often observed in dogs, followed by cats, iguanas, ferrets, horses, and cockatoos (Fig 2A–inset). The number of toxicosis cases reported among all surveyed veterinarians was significantly higher after October 2018 (p<0.0001; Fig 2A). This trend was observed both in Canada (p<0.0001; Fig 2B) and in the US (p = 0.002; Fig 2C). In all hospital settings and practice types, the numbers of cannabis toxicosis cases were reported to be higher post-legalization (all p≤0.0001; Fig 2D–2I).

### Changes in reports of toxicosis before and after 2018

At the individual level, most veterinarians reported no difference in the number of cannabis toxicosis cases annually pre- and post-2018 (Fig 3A–3C). However, almost all veterinarians who reported changes between the two periods reported an increase in the number of cases they observed, in both Canada (Fig 3B) and the US (Fig 3C).

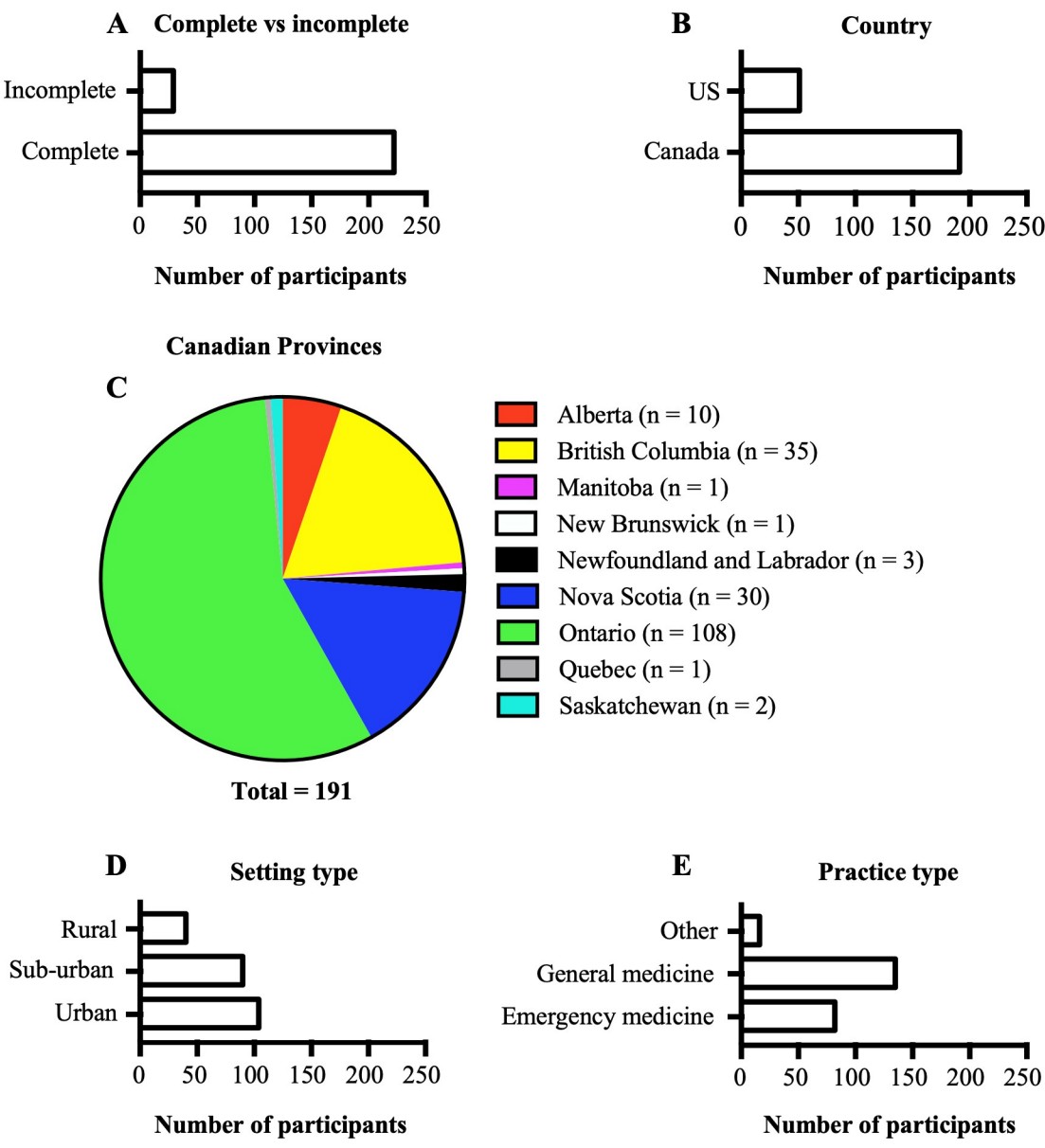

**Fig 1. Demographics of participants.** A. Graph showing the number of participants who completed the survey and those who did not. B. Graph showing the countries in which the study participants practice. C. A pie chart showing the distribution of participants who practiced in Canada according to provinces. D. Graph showing the hospital setting in which participants practice. E. Graph showing the type of medicine practiced by participants.

## Commonly observed clinical signs

The clinical signs that veterinarians reported to have observed most commonly (in decreasing order) were: urinary incontinence, disorientation, ataxia, lethargy, hyperesthesia, bradycardia, stupor/obtundation, and twitching (Table 1). A small number of veterinarians reported witnessing other signs including head bobbing and hyperthermia. The Chi square Goodness of Fit test and the post-hoc binomial pairwise test revealed that urinary incontinence, disorientation, ataxia, lethargy, hyperesthesia, and bradycardia were the clinical signs that occurred most

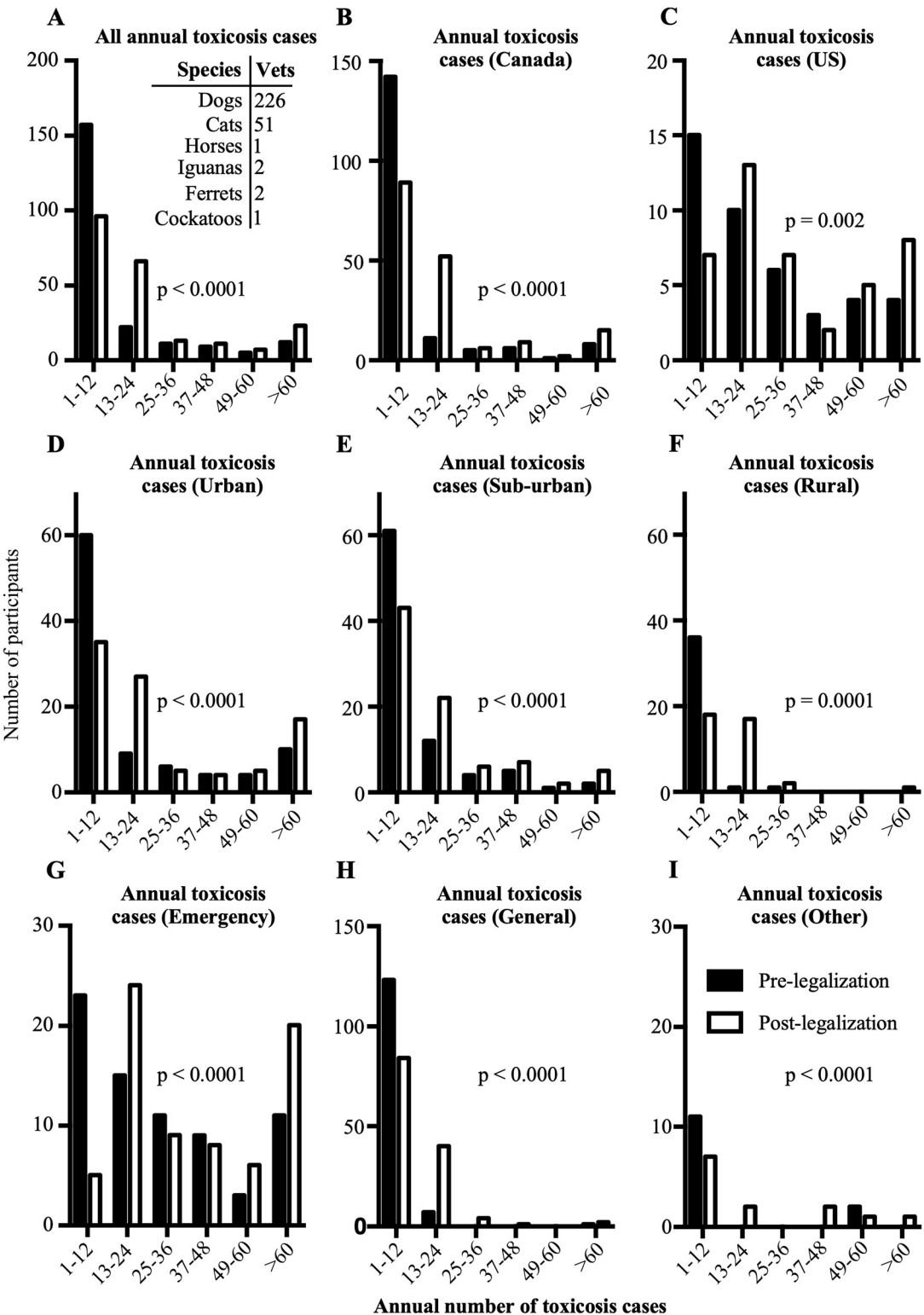

**Fig 2. Reported annual number of toxicosis cases before and after October 2018.** A. All reported annual cannabis toxicosis cases pre- and post-legalization. Inset: Species in which cannabis toxicosis was observed and the number of participants who reported them. B. Reported annual cannabis toxicosis cases pre- and post-legalization in Canada. C. Reported annual cannabis toxicosis cases before and after October 2018 in the US. D. Reported annual cannabis toxicosis cases pre- and post-legalization in urban settings. E. Reported annual cannabis toxicosis cases pre- and post-legalization in sub-urban settings. F. Reported

annual cannabis toxicosis cases pre- and post-legalization in rural settings. G. Reported annual cannabis toxicosis cases pre- and post-legalization by participants who practice emergency medicine. H. Reported annual cannabis toxicosis cases pre- and post-legalization by participants who practice general medicine. I. Reported annual cannabis toxicosis cases pre- and post-legalization by participants who practice other types of medicine. Vets: Number of veterinarians who reported being presented with cannabis toxicosis in a particular species.

frequently (Table 1). Interestingly, except for bradycardia, all these clinical signs were reported to be usually severe.

Following a Chi square Goodness-of-Fit test and a binomial pairwise test, the frequency category which was significantly different from the other frequency categories were identified and reported as well.

## Products and routes of exposure

As shown in Fig 4A, the products that often led to cannabis toxicosis in pets were edibles and dried cannabis. Other products reported by veterinarians to cause cannabis toxicosis were discarded joint butts, human feces, cannabis-infused butter/oil, and compost (Fig 4B). Most veterinarians (n = 105/196) reported that they or the pet owner did not know the source of cannabis exposure. However, among those who reported the sources of cannabis products that led to cannabis toxicosis (n = 101/196), most (n = 34/196) reported that they were obtained from government regulated producers, followed by home cultivated plants (n = 29/196), and the black market (n = 28/196) (results not shown). The most common route of exposure (Fig 4C) was ingestion, and ingestion while unattended was the most cited reason for exposure (Fig 4D).

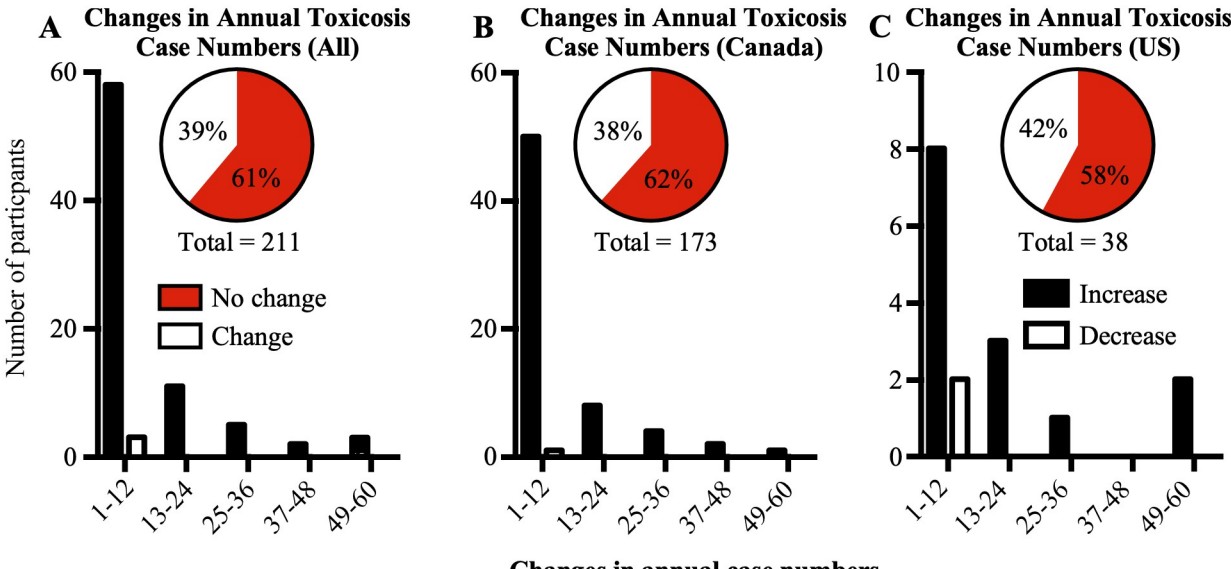

**Fig 3. Changes in annual cannabis toxicosis case numbers reported by each participant.** A. Graph showing changes in annual cannabis toxicosis case numbers reported by all participants. Insets: Pie chart showing number of participants who reported equal annual number of cases pre- and post-legalization (no change) and those who reported different annual numbers of cases pre- and post-legalization (change) B. Changes in annual cannabis toxicosis case numbers reported by participants in Canada. C. Changes in annual cannabis toxicosis case numbers reported by participants in the US. Increase: participants who reported increases in annual numbers of cannabis toxicosis cases pre- and post-legalization; Decrease: participants who reported decreases in annual numbers of cannabis toxicosis cases pre- and post-legalization. Inset pie chart: No change: participants who reported equal annual numbers of cannabis toxicosis cases pre- and post-legalization; Change: participants who reported different annual numbers of cannabis toxicosis cases pre- and post-legalization.

**Table 1. Aggregate data about cannabis toxicosis clinical signs and their frequencies reported by Canadian and American veterinarians.**

| Clinical Signs (No. veterinarians) | Frequencies | | | | Significant frequency |
|---|---|---|---|---|---|
| | **Very often** | **Often** | **Sometimes** | **Rare** | |
| Anorexia (16) | 2 | 8 | 5 | 1 | |
| Bradycardia (112) | 28 | 56 | 25 | 3 | Often |
| Disorientation (182) | 116 | 57 | 8 | 1 | Very often |
| Diarrhea (4) | 0 | 2 | 2 | 0 | |
| Hypertension (5) | 0 | 1 | 4 | 0 | |
| Hypotension (15) | 1 | 5 | 5 | 4 | |
| Increased anxiety (70) | 18 | 34 | 14 | 4 | |
| Dry mouth/ excessive drinking (11) | 1 | 1 | 9 | 0 | |
| Polyphagia (8) | 0 | 4 | 3 | 1 | |
| Vocalizing/Crying (27) | 3 | 13 | 11 | 0 | |
| Seizures (2) | 0 | 0 | 0 | 2 | |
| Tachycardia (19) | 0 | 7 | 9 | 3 | |
| Vomiting (57) | 4 | 19 | 25 | 9 | |
| Lateral recumbency (42) | 2 | 10 | 14 | 16 | |
| Urinary incontinence (195) | 131 | 54 | 6 | 1 | Very often |
| Lethargy (150) | 98 | 43 | 7 | 2 | Very often |
| Stupor/Obtundation (104) | 17 | 38 | 34 | 15 | |
| Hypothermia (60) | 4 | 31 | 22 | 3 | |
| Tremors (56) | 8 | 23 | 18 | 7 | |
| Agitation (64) | 12 | 28 | 21 | 3 | |
| Respiratory depression (13) | 0 | 6 | 5 | 2 | |
| Ataxia (178) | 131 | 41 | 4 | 2 | Very often |
| Mydriasis (78) | 24 | 44 | 10 | 0 | |
| Hyperesthesia (134) | 76 | 39 | 16 | 3 | Very often |
| Twitching (93) | 20 | 33 | 34 | 6 | |
| Ptyalism (44) | 3 | 17 | 23 | 1 | |

## Diagnosis and treatment

Pets that presented at veterinary hospitals were diagnosed with cannabis toxicosis based on supportive clinical signs, a history of possible/known exposure, and/or the use of over-the-counter urine drug tests (Fig 5A). Following diagnosis, the most common treatments included: outpatient monitoring and supportive care, administration of intravenous fluids, in-hospital monitoring only, administration of activated charcoal, induction of emesis, administration of anti-emetics, thermal support (warming/cooling), and blood pressure monitoring (Fig 5B). Animals were usually treated either as outpatients or they were hospitalized for less than 24 hours (Fig 5C). Most participants reported that all clinical signs resolved following cannabis exposure, except for a few pets that reportedly died in association with cannabis toxicosis (n = 16 animals). The cost of treatment for majority of the cases was less than CAD$ 500 (Fig 5D).

Even though most of the veterinarians reported no deaths (211/221), 10/221 veterinarians reported a total of 16 deaths believed to be attributable to cannabis toxicosis (Fig 5E). Other than euthanasia (n = 2), the causes of death reported to be associated with cannabis exposures were aspiration pneumonia (n = 5), respiratory arrest (n = 3), uncontrolled seizures (n = 2), coma (n = 2), and pancreatitis (n = 1) (Fig 5F).

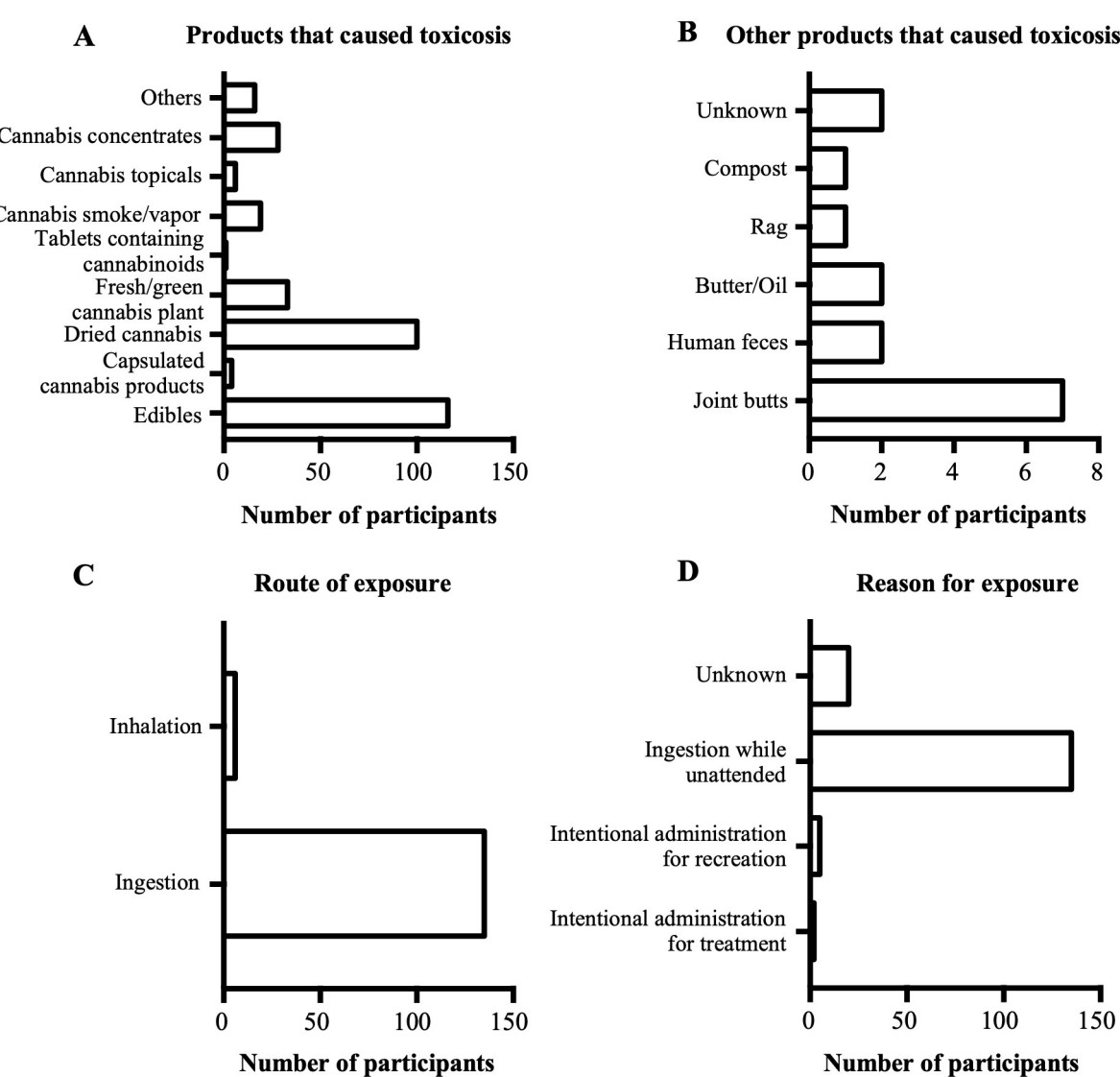

**Fig 4. Products that caused cannabis toxicosis, their routes of exposure, and reasons for exposure.** A. Graph showing the products that were reported to cause cannabis toxicosis by the study participants. B. Graph showing other products that were reported to cause cannabis toxicosis by the study participants. C. Graph showing the route of exposure to the products that caused cannabis toxicosis. D. Graph showing the reasons for exposure to the products that caused cannabis toxicosis.

## Discussion

Cannabis toxicosis was frequently reported in dogs, and in both Canada and the US, the number of cannabis toxicosis cases increased significantly after October 2018 (which coincided with legalization in Canada, but not the US). Additionally, of those who reported a change (85/211), nearly all (82/85) reported an increase in the number of cases. Among the reported clinical signs of cannabis toxicosis (primarily observed in dogs, therefore clinical signs we report herein are likely biased towards canine-specific presentations), urinary incontinence, ataxia, disorientation, bradycardia, hyperesthesia, and lethargy were most common. The product which often caused cannabis toxicosis was edibles, and the most common route of exposure was via oral ingestion, with the most common reason being ingestion while unattended.

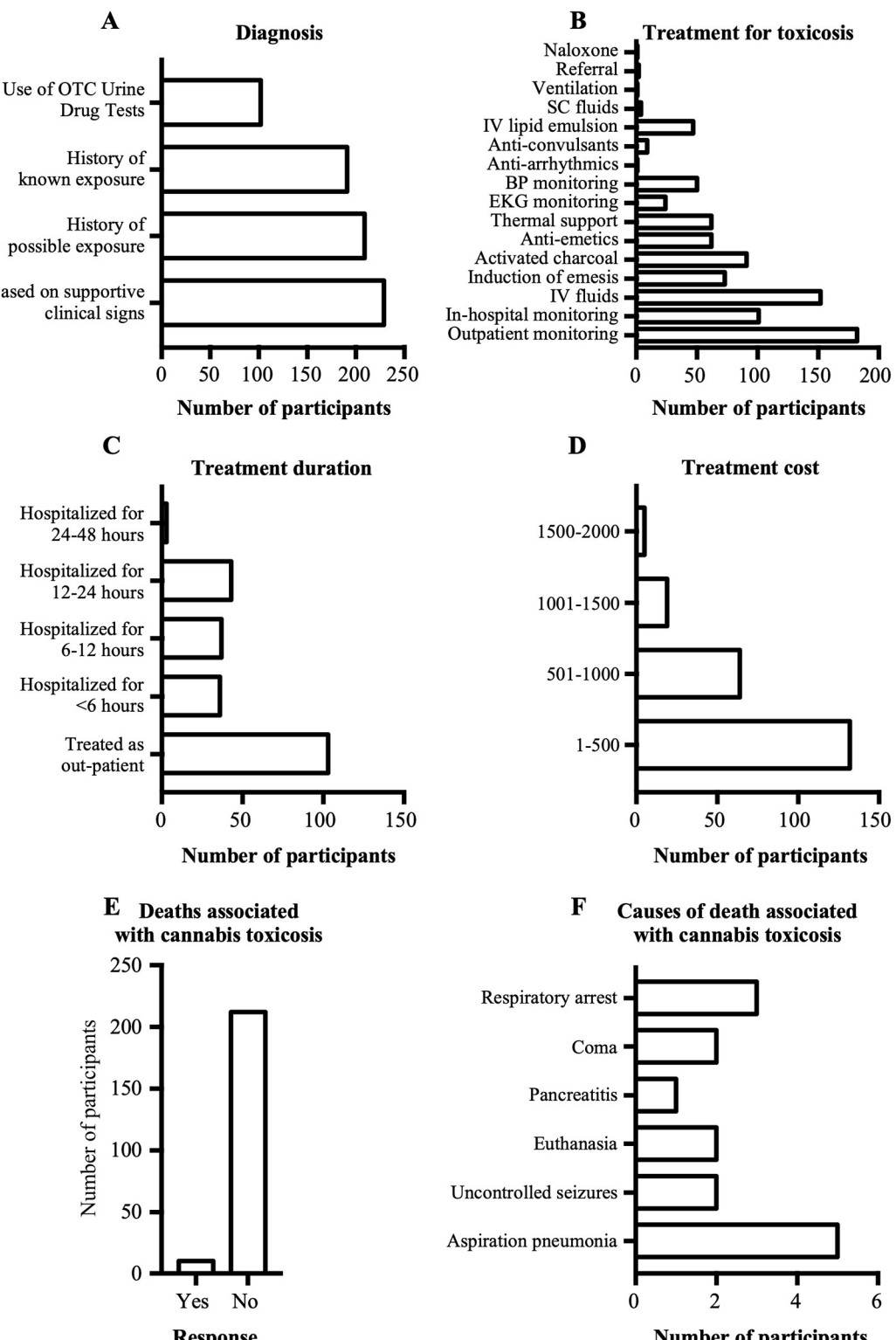

**Fig 5. Diagnosis and treatments for cannabis toxicosis, and deaths associated with cannabis toxicosis and their causes.**
A. Graph showing methods used by participants to diagnose cannabis toxicosis. B. Graph showing various treatments for cannabis toxicosis reported by participants. C. Graph showing treatment duration following cannabis toxicosis. D. Graph showing cost of treatment for cannabis toxicosis. E. Graph showing the number of participants who either reported deaths or no deaths associated with cannabis toxicosis in pets. F. Graph showing the causes of deaths reportedly associated with cannabis toxicosis.

Diagnosis was frequently based on the presence of supportive clinical signs, and the most common treatment was outpatient monitoring, which lasted for less than 48 hours. Except for a few patients that were reported to have died in association with cannabis exposure, all patients recovered completely after treatment, with a total treatment cost less than CAD $500.

Similar to other studies [9], the pets that were treated most often by the veterinarians in our sample were cats and dogs. This is consistent with a recent survey which revealed that there were 7.7 million dogs and 8.1 million cats in Canadian households [10]. In our study, cannabis toxicoses were frequently observed in dogs compared to cats, similar to that previously reported [9]. Consistent with previous work, participants also reported cannabis toxicoses in other companion animal species such as horses, ferrets, and iguanas [9, 11, 12], and also in previously unreported species such as pet cockatoos.

Similar to a number of previous studies [5, 13], we observed an overall increase in the number of cannabis toxicosis cases after October 2018, even though our analysis at the participant-level revealed that majority of the participants reported equal case numbers pre- and post-legalization. This could be because participants did not report the actual case numbers but rather selected among predetermined numeric ranges. As such, small increases that were within the same range, would have been reported as "no change." The increase in case numbers could be due to any combination of the following factors: 1. legalization of cannabis for medical and recreational use in Canada; 2. increased reporting by pet owners due to legalization; and 3. increased awareness of veterinarians about cannabis toxicoses [5, 13]. Moreover, it is important to consider the pharmacology of cannabinoids in cannabis, and how they may have contributed to the findings noted above.

## Pharmacokinetics and pharmacodynamics of THC

The two major cannabinoids in cannabis are delta-9-tetrahydrocannabinol (THC) and its isomer, cannabidiol (CBD) [14]. THC is the main psychoactive cannabinoid that produces euphoria but can also have intoxicating effects [14, 15]. Meanwhile, CBD is considered non-psychoactive despite its anxiolytic, antipsychotic, and anti-inflammatory effects [16–18]. Despite the difficulty in determining the exact time to onset of clinical signs following cannabis toxicosis since owners only seek medical attention upon the appearance of clinical signs; in dogs, the onset of clinical signs ranges from within minutes post-inhalation [19] to several hours post-ingestion [20, 21]. This delay in onset may be due to the long biological half-life of THC in dogs due to adipose tissue storage, even though the THC plasma half-life is relatively short [19]. The delayed onset of clinical signs may also be due to the time it takes for THC to undergo first-pass liver metabolism post-ingestion [20]. When compared to humans, dogs seem to have similar oral absorption, but a much longer duration and wider range of clinical signs. Dogs produce the additional THC metabolites 8-OH-$\Delta^9$-THC and 11-OH-THC, which may contribute to the additional clinical signs observed only in dogs [19, 22]. 11-OH-THC is an active metabolite that may be produced in larger quantities after cannabis ingestion following first pass metabolism.

Given that dogs were the species in which cannabis toxicosis was reported most frequently in our study, and that majority of the signs of cannabis toxicosis were neurological, similar to what others have reported in dogs, [21], the subsequent discussion will predominantly focus on these neurological signs. Among the clinical signs of cannabis toxicosis reported in our study, the most common was urinary incontinence. Expression of cannabinoid receptors has been demonstrated in the bladders of humans, rats, and mice [23–25]. Even though the mechanism by which THC regulates bladder contractility *in vivo* is unclear, it was previously shown that in mice, the administration of THC in bladder tissue inhibited electrically-evoked

contractions of the bladder [24], which could lead to urinary incontinence. This was also demonstrated in rats using a CB1 receptor agonist [23]. Cannabinoid receptors have not yet been confirmed in the dog bladder; however, if cannabinoid receptor expression is conserved, bladder hyperactivity through increases in contraction could be contributing to the reported incontinence.

Cannabis toxicosis also resulted in bradycardia. In rats, the bradycardic effect of THC on the heart involves CB1-like cannabinoid receptors [26]. Although a previous study has shown that bradycardia may result from the effects of THC directly on catecholamine receptors (including adrenergic receptors) in the heart [27], another study in cats concluded that THC decreases central adrenergic neuronal activity, leading to decreased sympathetic tone, and subsequently causing bradycardia [28]. Schmid, Schwartz [29] reported the presence of the endocannabinoids, anandamide and 2-arachidonoylglycerol in the rat heart, while both CB1 and CB2 receptors were detected in the myocardia of rats, mice, and guinea pigs [30–32]. CB1 mRNA was also detected in the human heart [33, 34]. All the aforementioned studies suggest that the bradycardic effects of THC may be mediated by several types of receptors, including cannabinoid receptors, in the heart.

Due to the lipophilic nature of THC, it is easily taken up by highly-perfused organs like the brain [35], which may explain the neurological signs of cannabis toxicosis. Ataxia, a common neurological symptom of cannabis toxicosis in both humans and animals, refers to the lack of coordination during movement. In the brain, the region responsible for coordination is the cerebellum [36], which contains a higher number of CB1 receptors in dogs as compared to humans [37]. Patel and Hillard [38] showed that intraperitoneal THC administration in a mouse caused motor deficits including ataxia. They subsequently proposed that even though THC inhibits both excitatory and inhibitory synapses in the cerebellum, its main mechanism of action is the inhibition of the inhibitory synapses (basket cell/Purkinje cell), leading to the disinhibition of Purkinje cells. Firing of these GABAergic cells inhibits deep cerebellar nuclei cells, thus resulting in ataxia. A similar mechanism may occur in dogs, but further investigation is required.

Previous studies in chronic fatigue syndrome patients revealed that brain regions implicated include the basal ganglia, anterior cingulate, and frontal, temporal, and parietal regions [39, 40]. These regions also overlap with brain regions implicated in motivation [41]. Interestingly, these regions in humans and animals contain CB1 receptors [37, 42], which can be modulated by THC. This may explain how cannabis toxicosis can cause lethargy. Disorientation is the loss of a sense of direction and mental confusion. Frontal and temporal cortices, regions shown to contain CB1 receptors [37, 42, 43], are involved during mental orientation in space and time; the disorientation exhibited by pets during cannabis toxicosis may involve THC modulation of CB1 receptors in similar regions of the brain [44].

## Products that caused cannabis toxicosis and the routes and reasons for exposure

In our study, edibles were the most common cannabis product that resulted in toxicosis, which is not surprising since they are the most common form of cannabis products purchased for dogs [3]; however, it is difficult to ascertain from our findings whether these edible products were purchased for human or animal consumption. Pets are often exposed to homemade or commercial edible goods, which are typically made using THC butter [5]. In our study, and previous studies [11, 21], plant materials, including dried and fresh green cannabis, was another common product that led to cannabis toxicosis. The least common cannabis toxicosis-causing products were topical cannabis products, capsulated cannabis products, and tablets

containing cannabinoids. It is interesting that we captured human feces ingestion as a source of cannabis toxicosis in pets in North America, since a recent study in Melbourne, Australia reported similar findings in dogs and suggested the need for pet owners to use adequate hygiene measures when managing pets, especially outdoors [45].

The most common source of cannabis toxicosis-causing products reported by veterinarians was government-regulated producers, followed by home-cultivated plants. A few pet owners reported that they obtained the products from the black market, however, this might be susceptible to under-reporting. In our study, ingestion was reported as the most common route of exposure. Compared to inhalation [46], which was the second most common route of exposure reported in our study, ingestion of edibles made with THC butter has been reported to result in more severe clinical signs and a higher risk for cannabis toxicosis in animals, since a majority of animals presenting with moderate to severe clinical signs of toxicosis had ingested some form of cannabis product including edible goods [5, 11, 21]. Furthermore, the presence of other toxins (e.g., chocolate) in the edible product may have contributed to clinical illness, and may explain some of the deaths reported by veterinarians in our study and in another retrospective study [5]. Even though less common, the ingestion of synthetic cannabinoids also leads to more severe clinical signs [47], and is known to be potentially lethal in dogs [48, 49].

The most commonly stated reason for pet exposure to cannabis was via oral ingestion while unattended, which was also reported in a previous study [46], followed by intentional administration for recreation (given to pets for fun?), or as a medical treatment. Our findings suggest that pet owners would have to put measures in place to prevent pets from accessing cannabis products including restricting cannabis to hard-to-access areas of the house, putting their cannabis products in pet-proof containers, and monitoring pets when cannabis-based products may be accessible. Some pet owners stated that cannabis toxicosis occurred following medical treatment which may be a result of unintended over-administration of these drugs due to the delay in manifestation of their effects. A small number of participants reported that some pets, specifically dogs, were exposed while being walked. Another possibility, which was not captured in this study, is cannabis users exposing pets to cannabis products while they were themselves intoxicated with cannabis.

## Diagnosis of cannabis toxicosis

In our study, the most common diagnostic method was the use of supportive clinical signs, along with a history of possible/known exposure, and/or the use of over-the-counter urine drug test kits. The key to appropriate treatment and successful recovery is accurate diagnosis, based on clinical signs, and accurate medical history from pet owners. Pet owners may be inclined to withhold information from veterinarians regarding accidental exposures to drugs [21] for fear of legal consequences. Therefore, veterinarians must encourage owners to provide complete histories when possible [20].

Many veterinarians in our study reported diagnosing cannabis toxicosis using urine drug test kits. The use of urine test kits in dogs may be unreliable based on interactions with other drugs, since patients on nonsteroidal anti-inflammatory drugs could have false-positive results [50]. The incidence of false-negative results using the human urine drug test kit is also a concern. False negatives may occur if the urine sample is tested too soon after exposure [5], if the urine sample is not handled appropriately leading to the THC binding to the rubber stoppers and glass containers [5], if the patient consumed synthetic cannabinoids [51], or if the patient has diluted urine [52]. In dogs, false negatives can also occur since THC is metabolized into 8-OH-$\Delta^9$-THC, which may not be detected by the human urine drug test kits [53] since they were not designed to detect this compound.

## Treatments for cannabis toxicosis

In our study, the treatment method used most frequently was outpatient monitoring, followed by the administration of intravenous fluids, activated charcoal, and anti-emetics. Intravenous fluids can be administered as a form of supportive care [19, 47] to prevent both dehydration (i.e., from vomiting) and hypothermia [20] during cannabis toxicosis. Activated charcoal is often administered to prevent further absorption of the ingested material in the stomach and aid in decontamination [19]. This method was recommended in previous studies for many of the dogs that experienced cannabis toxicosis and for all the iguanas [5, 11, 21]. Induction of emesis is commonly performed in dogs, cats, and iguanas as an initial treatment if a toxic dose was ingested within 15–30 minutes or a significant amount of plant material remains in the stomach [5, 11, 19, 21, 46, 47, 54]. It is safest to perform this procedure if the patient is still asymptomatic and with a normal mentation, to decrease the risk of aspiration [19]. Emesis should never be induced if the animal is extremely agitated, severely depressed, or unresponsive [20]. The administration of intravenous lipid emulsion was a treatment method reported by several participants in our study. This method was reportedly used to treat a Boxer dog that ingested synthetic cannabinoids and was also used in a dog that died during treatment after ingesting THC butter, even though this method can have adverse effects such as leading to serum lipemia [5, 47]. However, it may be useful for patients that are unresponsive to conventional treatments [19].

Treatment duration and recovery time following cannabis toxicosis depends on the severity of the toxicosis, which is dependent on the dose of THC (or other cannabinoids), quantity of cannabis or cannabis products consumed, and the route of exposure. In our study, veterinarians reported that most animals were treated as outpatients, while the remaining patients were hospitalized for less than 48 hours. This is not surprising since pets usually recover within 72 hours after cannabis toxicosis [9, 19]. A wide range of recovery times have been reported in the literature, but they appear to vary between species [11, 21, 46, 47, 54].

## Potential lethality of cannabis requires further investigation

Although most of the cannabis toxicosis cases in companion animals made a full recovery, 10 veterinarians cited death as an outcome for 16 cases. The details surrounding each case were not captured, thus we cannot be certain that exposure to cannabis directly resulted in mortality, or that the presence of other toxins found in edible products (e.g., chocolate, xylitol), or other underlying medical conditions contributed to the fatalities. In certain cases, it appears that cannabis was unlikely to be the primary cause of death, such as with aspiration pneumonia. In other cases, it may be possible that cannabis may have resulted in death directly, for example cases that report coma, uncontrolled seizures, or respiratory arrest as the primary clinical signs. These clinical signs are consistent with the mechanism of lethality in rats as reported by Thompson, Rosenkrantz [55], but the lethality of cannabis in dogs has not yet been confirmed. Previous research aiming to determine the lethal dose of cannabis in dogs was unable to determine a lethal dose (administering up to 5000 mg/kg of crude cannabis extract or 3000 mg/kg of delta-9-tetrahydrocannabinol or delta-8-tetrahydrocannabinol orally), and this issue has been the subject of controversy in the veterinary field, with several sources misreferencing this original scientific study [56, 57]. Previous field reports claiming that cannabis resulted in the deaths of animals arrived at this diagnosis through exclusion of other diagnoses [5], and thus do not represent strong scientific evidence; further basic research is needed to determine the potential lethality of non-synthetic cannabis in dogs and other pets, and its mechanism, if applicable. The suspected cases documented here, however, provide some guidance regarding this research gap; small and/or young animals may be more likely to

be exposed to a higher apparent dosage, particularly for cannabis edibles, and due to their small body mass, could theoretically be more likely to succumb to an overdose and associated central nervous system depression, as was seen in rats in the lethality study [55].

Regardless of lethality, aggressive treatment of young and/or small animals is warranted in most cases, since the dosage may be unknown, and decontamination with emetics, IV fluids, and activated charcoal is considered a relatively safe treatment course. Naloxone infusions may also be considered in severe cannabis toxicoses cases, since there is some clinical evidence from human medicine that this opioid antagonist is effective in treating cannabis overdoses, because it also binds to endocannabinoid receptors [58].

## Limitations

The aggregate data collected by this veterinarian-based survey are prone to several biases. Since the survey was voluntary, a selection bias could have skewed the data; participants from states or provinces where cannabis is legal for recreational use may be more likely to see or report cases of cannabis toxicosis in animals compared with participants practicing in states or provinces where cannabis remains illegal for recreational use in humans. Furthermore, this survey data may be prone to recall bias, as veterinarians may not accurately remember the details of previous cases. However, from the specific outcomes reported, they might have consulted their records while completing the survey. Most importantly, the type of data collected here represents subjective aggregated data concerning cannabis toxicosis cases seen and reported by veterinarians; thus, raw numerical data concerning individual animals was not captured here. Consequently, the data presented herein should be interpreted with caution, and are, in some cases, inevitably vague and imprecise, particularly for the types and frequencies of clinical signs. Additionally, our data may also be prone to misclassification bias because of the lack of highly sensitive and specific diagnostic tests to confirm cannabis intoxication in animals. Thus, most of the diagnoses were made based on clinical signs along with a history of possible or known exposure. The latter requires veterinarians to rely on the history reported by pet owners, which may not always be completely honest due to the stigma which continues to surround cannabis, despite legalization.

## Conclusions

Based on our veterinarian-reported survey data, the incidence of cannabis toxicoses in companion animals (primarily dogs) appears to have increased following legalization of cannabis for recreational purposes in Canada in October 2018. Although several factors may account for this apparent increase in cannabis toxicosis cases, the increased availability of cannabis products for humans is likely an important factor, since most of the toxicoses reported here resulted from inadvertent exposures; however, edibles were not legalized in Canada until October 2019, even though edibles were reported as the most common source of exposure in our study. The lack of veterinary oversight regarding the medicinal use of cannabis for animals in Canada also remains problematic and may also be contributing to a certain portion of these reported toxicoses, as many pet owners attempt to self-medicate their animals with these products (some of which are from the black market). Most of the cannabis toxicoses in animals appear to be benign; most cases resulted in mild to moderate clinical signs (most commonly, lethargy, disorientation, urinary incontinence, ataxia, and hyperesthesia), were treated as outpatients, and nearly all animals were reported to have fully recovered. Although several veterinarians in our survey reported deaths in association with cannabis exposure, rigorous controlled laboratory studies are needed to investigate this important and controversial issue, to eliminate or control for the presence of confounders such as other toxins (e.g., illicit drugs,

chocolate, xylitol), other underlying disease processes, or causes of death secondary to cannabis ingestion (e.g., aspiration pneumonia). Finally, the use of clinical history and over-the-counter urine drug tests, although routinely used to diagnose cannabis toxicity cases in clinical practice, may be prone to false positive or false negative test results. There is a need for more sensitive and specific diagnostic tests to diagnose cannabis toxicities, whether to support aggressive decontamination procedures in high-risk patients, or to differentiate between non-synthetic cannabis (lethality unknown) and synthetic cannabis (known to be lethal in dogs; Hanasono, Sullivan [48]). As the burgeoning field of medicinal cannabis use in humans and animals continues to grow, fundamental research into the pharmacokinetics, pharmacodynamics, and potential lethality of cannabis in different animal species is also needed to address outstanding research gaps.

## Supporting information

**S1 File. Sample of online survey questionnaire.**
(PDF)

**S1 Table. Spreadsheet containing minimal data set.**
(XLSX)

## Acknowledgments

We would like to thank the statisticians at the University of Guelph Library who assisted us with statistical analyses and with using Qualtrics.

## Author Contributions

**Conceptualization:** Richard Quansah Amissah, Karolina Urban, Jibran Khokhar.

**Data curation:** Richard Quansah Amissah, Nadine A. Vogt, Chuyun Chen.

**Formal analysis:** Richard Quansah Amissah, Nadine A. Vogt.

**Funding acquisition:** Karolina Urban, Jibran Khokhar.

**Investigation:** Richard Quansah Amissah.

**Methodology:** Richard Quansah Amissah, Nadine A. Vogt.

**Project administration:** Richard Quansah Amissah, Jibran Khokhar.

**Supervision:** Richard Quansah Amissah.

**Validation:** Richard Quansah Amissah.

**Visualization:** Richard Quansah Amissah.

**Writing – original draft:** Richard Quansah Amissah, Nadine A. Vogt, Chuyun Chen.

**Writing – review & editing:** Richard Quansah Amissah, Nadine A. Vogt, Chuyun Chen, Karolina Urban, Jibran Khokhar.

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
