## [Decision Letter · Decision Letter 0]

24 Jan 2022

PONE-D-21-38974Prevalence and Characteristics of Cannabis-induced Toxicoses in Pets: Results from a Survey of Veterinarians in North AmericaPLOS ONE

Dear Dr. Khokhar,

Thank you for submitting your manuscript to PLOS ONE. After careful consideration, we feel that it has merit but does not fully meet PLOS ONE’s publication criteria as it currently stands. Therefore, we invite you to submit a revised version of the manuscript that addresses the points raised during the review process.

We look forward to receiving your revised manuscript.

Kind regards,

Benito Soto-Blanco, DVM, MSc, PhD

Academic Editor

PLOS ONE

Journal Requirements:

"This research was funded by a Natural Sciences and Engineering Research Council Alliance Grant (ALLRP 549529 to JYK; https://www.nserc-crsng.gc.ca/innovate-innover/alliance-alliance/index_eng.asp) and a MITACS Accelerate Fellowship (IT27597 to RQA and JYK; https://www.mitacs.ca/en/programs/accelerate/fellowship) in partnership with Avicanna Inc."

"This research was funded by a Natural Sciences and Engineering Research Council Alliance Grant (ALLRP 549529 to JYK; https://www.nserc-crsng.gc.ca/innovate-innover/alliance-alliance/index_eng.asp) and a MITACS Accelerate Fellowship (IT27597 to RQA and JYK; https://www.mitacs.ca/en/programs/accelerate/fellowship) in partnership with Avicanna Inc."

We note that you have provided funding information. However, funding information should not appear in the Funding section or other areas of your manuscript. We will only publish funding information present in the Funding Statement section of the online submission form. 

"This research was funded by a Natural Sciences and Engineering Research Council Alliance Grant (ALLRP 549529 to JYK; https://www.nserc-crsng.gc.ca/innovate-innover/alliance-alliance/index_eng.asp) and a MITACS Accelerate Fellowship (IT27597 to RQA and JYK; https://www.mitacs.ca/en/programs/accelerate/fellowship) in partnership with Avicanna Inc." 

"Dr. Urban is an employee of Avicanna Inc., during which time she has received stock options. Avicanna Inc. did not influence the design, conduct or interpretation of the data derived from this study."

We note that you received funding from a commercial source: Avicanna Inc.

Within this Competing Interests Statement, please confirm that this does not alter your adherence to all PLOS ONE policies on sharing data and materials by including the following statement: ""This does not alter our adherence to PLOS ONE policies on sharing data and materials.” (as detailed online in our guide for authors http://journals.plos.org/plosone/s/competing-interests).  If there are restrictions on sharing of data and/or materials, please state these. Please note that we cannot proceed with consideration of your article until this information has been declared. 

Reviewers' comments:

Reviewer's Responses to Questions

**Comments to the Author**

1. Is the manuscript technically sound, and do the data support the conclusions?

Reviewer #1: Yes

Reviewer #2: Yes

2. Has the statistical analysis been performed appropriately and rigorously? 

Reviewer #1: Yes

Reviewer #2: Yes

3. Have the authors made all data underlying the findings in their manuscript fully available?

Reviewer #1: No

Reviewer #2: Yes

4. Is the manuscript presented in an intelligible fashion and written in standard English?

Reviewer #1: Yes

Reviewer #2: Yes

5. Review Comments to the Author

Reviewer #1: This is very interesting, but I am unclear about a few points. I could not see the questionnaire so the information may be there, but it would be useful in the text. I need more information to fully understand the data included int he study.

What is meant by cannabis here? Does it include CBD products and synthetic cannabinoid receptor agonists (SCRAs)? So, both illicit drugs and medical products for human and veterinary use? Mixed exposure cases were included or excluded (e.g. chocolate, xylitol)?

I do not really understand how the numbers were derived. What were respondents asked about the number of cases? Was it the annual cases they saw or the number of cases between a particular period? This is important as you have numbers before and after legalisation in Canada. Were they answering from memory or from their practice case database?

What were the signs seen in the animals other than cats or dogs? Can this be extracted? This could be useful, particularly in animals not previously reported with cannabis exposure. How were these non dog/cat cases diagnosed? Would you suspect cannabis in a drowsy ferret, for example? Or were drug urine tests or owner history available in these cases?

Line 255. Cannabis poisoning has been reported in a ferret

Smith RA. Coma in a ferret after ingestion of cannabis. Vet Hum Toxicol. 1988 Oct;30(5):486.

Figure 3. I do not understand these figures. Over what period are these changes in case numbers? Is the decrease in US cases in 3c an effect of the different timing of legalisation?

Line 412. Were all the fatal cases in dogs?

Line 353. I would say SRACs are ‘known to be potentially lethal in dogs’, as not all cases are fatal and there are hundreds of SCRAs.

Van-de-Velde M, Bates N, Edwards N. Suspected synthetic cannabinoid ingestion in dogs: a case series [abstract]. Clinical Toxicology. 2017;55(5):44-45. https://www.eapcct.org/publicfile.php?folder=congress&file=Abstracts_Basel.pdf

Sioris KM, Keyler D. 2014 K2 and K9s [abstract]. Clin Toxicol 52:710.

https://www.eapcct.org/publicfile.php?folder=congress&file=Abstracts_NewOrleans2.pdf

Line 356. Users may give cannabis products to pets while they are under the influence.

Line 389. Was repeat dose activated charcoal used in these cases or single dose activated charcoal.

Line 398. ’Intra-lipids’ is a trade name (Intralipid®) so would be better to use intravenous lipid emulsion (IVLE).

Figure 4. It is interesting you mention human faeces as a source of exposure. This paper has just been published.

Lauinger CA, Peacock R. Marijuana toxicosis in dogs in Melbourne, Australia, following suspected ingestion of human faeces: 15 cases (2011-2020). Aust Vet J. 2021 Nov 16. doi: 10.1111/avj.13134. Epub ahead of print.

Reviewer #2: This is a very interesting paper concerning a type of intoxication that seems to have some incidence in Canada and USA, with an increase after legalization of cannabis use in these countries. This can be relevant for other countries where this legalization occurs.

The information about clinical signs, products and route of exposure and diagnosis and treatment of the intoxication is also important and well reported and discuss.

The study have some limitation that were identified by the authors.

Only graphs from figure 2 and 3 are not so well done because x-axis is not explicit. Can the authors try to better explain these data?

6. PLOS authors have the option to publish the peer review history of their article (what does this mean?). If published, this will include your full peer review and any attached files.

Reviewer #1: No

Reviewer #2: No

---

## [Author Response · Author response to Decision Letter 0]

8 Mar 2022

Dear Editor,

We thank the reviewers and the editorial team for their constructive comments and feedback. Below, we have responded to the reviews provided by each referee, and we have correspondingly updated the manuscript for resubmission. We believe these changes have significantly improved the manuscript and we hope that it will be deemed suitable for publication in PLOS ONE. We will continue to build upon the findings from this manuscript, and look forward to incorporating the suggestions from the reviewers in future studies. 

We confirm that neither the manuscript nor any parts of its content are currently under consideration or published in another journal. All authors have approved the manuscript and agree with its submission to PLOS ONE.

Sincerely, 

Jibran Khokhar

Responses to Reviewers:

Reviewer 1

Thank you to the reviewers for their continued insight and help with our manuscript. We have addressed the comments below. 

1. This is very interesting, but I am unclear about a few points. I could not see the questionnaire so the information may be there, but it would be useful in the text. I need more information to fully understand the data included in the study. What is meant by cannabis here? Does it include CBD products and synthetic cannabinoid receptor agonists (SCRAs)? So, both illicit drugs and medical products for human and veterinary use? Mixed exposure cases were included or excluded (e.g. chocolate, xylitol)?

We have added a definition for what we refer to as cannabis to the text on lines 84 – 85: “In this study, cannabis refers to substances derived from either the cannabis plant or synthetic cannabinoids.” We did not collect data on whether the cannabis was illicit or for medical purposes for human or veterinary use, therefore, the data may include both types of cannabis products. We also pointed this out on lines 363 – 364 as follows: “however, it is difficult to ascertain from our findings whether these edible products were purchased for human or animal consumption.”

We did not distinguish cases based on toxins present in the cannabis product consumed since the data we collected did not contain this information. We therefore discussed this issue in the discussion section of the paper on lines 381 – 384, 450 – 453, and 510 – 515.

2. I do not really understand how the numbers were derived. What were respondents asked about the number of cases? Was it the annual cases they saw or the number of cases between a particular period? This is important as you have numbers before and after legalization in Canada. Were they answering from memory or from their practice case database?

Question #8 in the questionnaire was: “How many "suspected or confirmed" cannabis toxicosis cases were you presented with yearly as the primary veterinarian before cannabis legalization in Canada (October 17, 2018)?” and Question #9 was: “How many "suspected or confirmed" cannabis toxicosis cases are you presented with yearly as the primary veterinarian following cannabis legalization in Canada (October 17, 2018)?” For both questions, participants were given the option to choose among pre-defined ranges (1 – 12, 13 – 24, 25 – 36, 37 – 48, 49 – 60, and >60).

The numbers were annual number of cases for two periods: 1. before recreational cannabis use legalization and 2. after recreational cannabis use legalization. In Fig 2, we have modified the x-axis from “Number of toxicosis cases” to “Annual number of toxicosis cases” to improve clarity. We have also added the word “annual” to the title of Fig 2 and the legend for clarity as follows:

“Fig 2. Reported annual number of toxicosis cases before and after October 2018. A. All reported annual cannabis toxicosis cases pre- and post-legalization. Inset: Species in which cannabis toxicosis was observed and the number of participants who reported them. B. Reported annual cannabis toxicosis cases pre- and post-legalization in Canada. C. Reported annual cannabis toxicosis cases before and after October 2018 in the US. D. Reported annual cannabis toxicosis cases pre- and post-legalization in urban settings. E. Reported annual cannabis toxicosis cases pre- and post-legalization in sub-urban settings. F. Reported annual cannabis toxicosis cases pre- and post-legalization in rural settings. G. Reported annual cannabis toxicosis cases pre- and post-legalization by participants who practice emergency medicine. H. Reported annual cannabis toxicosis cases pre- and post-legalization by participants who practice general medicine. I. Reported annual cannabis toxicosis cases pre- and post-legalization by participants who practice other types of medicine. Vets: Number of veterinarians who reported being presented with cannabis toxicosis in a particular species.”

We discussed the issue related to recall bias in the limitations section of the paper on lines 482 – 484: “Furthermore, this survey data may be prone to recall bias, as veterinarians may not accurately remember the details of previous cases.” We also added the following sentence to lines 484 – 485: “However, from the specific outcomes reported, they might have consulted their records while completing the survey.”

3. What were the signs seen in the animals other than cats or dogs? Can this be extracted? This could be useful, particularly in animals not previously reported with cannabis exposure. How were these non-dog/cat cases diagnosed? Would you suspect cannabis in a drowsy ferret, for example? Or were drug urine tests or owner history available in these cases?

For brevity, and in order not to discourage participants from completing the survey, we did not inquire about the details surrounding the clinical signs presented by pets and the methods used for diagnosis according to species of pets. However, these questions could be answered with more comprehensive surveys in future studies.

4. Line 255. Cannabis poisoning has been reported in a ferret

Smith RA. Coma in a ferret after ingestion of cannabis. Vet Hum Toxicol. 1988 Oct;30(5):486.

We have revised the sentence and added the reference you suggested to lines 281 - 283 as follows: “Consistent with previous work, participants also reported cannabis toxicoses in other companion animal species such as horses, ferrets, and iguanas (7, 9, 10), and also in previously unreported species such as pet cockatoos.”

5. Figure 3. I do not understand these figures. Over what period are these changes in case numbers? Is the decrease in US cases in 3c an effect of the different timing of legalization?

We have revised the titles of the graphs in Fig 3 from “Changes in Toxicosis Case Numbers” to “Changes in Annual Toxicosis Case Numbers” for clarity. We have also added the word “annual” to the legend title and the legend for Fig 3 as follows:

“Fig 3. Changes in annual cannabis toxicosis case numbers reported by each participant. A. Graph showing changes in annual cannabis toxicosis case numbers reported by all participants. Insets: Pie chart showing number of participants who reported equal annual number of cases pre- and post-legalization (no change) and those who reported different annual numbers of cases pre- and post-legalization (change) B. Changes in annual cannabis toxicosis case numbers reported by participants in Canada. C. Changes in annual cannabis toxicosis case numbers reported by participants in the US. Increase: participants who reported increases in annual numbers of cannabis toxicosis cases pre- and post-legalization; Decrease: participants who reported decreases in annual numbers of cannabis toxicosis cases pre- and post-legalization. Inset pie chart: No change: participants who reported equal annual numbers of cannabis toxicosis cases pre- and post-legalization; Change: participants who reported different annual numbers of cannabis toxicosis cases pre- and post-legalization.”

The decrease in US cases could be due to the different timing of legalization across states and when compared to Canada, since several studies have reported increases in number of cannabis toxicosis cases in pets in the US.

6. Line 412. Were all the fatal cases in dogs?

As we did not ask participants about the species of pets in which deaths occurred, we can only speculate that since most of the toxicosis cases were reported in dogs, it also likely that most of the fatal cases were in dogs as well; however, future studies will be required to confirm this.

7. Line 353. I would say SRACs are ‘known to be potentially lethal in dogs’, as not all cases are fatal and there are hundreds of SCRAs.

Van-de-Velde M, Bates N, Edwards N. Suspected synthetic cannabinoid ingestion in dogs: a case series [abstract]. Clinical Toxicology. 2017;55(5):44-45. https://www.eapcct.org/publicfile.php?folder=congress&file=Abstracts_Basel.pdf

Sioris KM, Keyler D. 2014 K2 and K9s [abstract]. Clin Toxicol 52:710.

https://www.eapcct.org/publicfile.php?folder=congress&file=Abstracts_NewOrleans2.pdf

We have modified the sentence and cited the references that you suggested online 385 as follows: “Even though less common, the ingestion of synthetic cannabinoids also leads to more severe clinical signs (45), and is known to be potentially lethal in dogs (46, 47).”

8. Line 356. Users may give cannabis products to pets while they are under the influence.

We have added a sentence to the text on lines 395 – 397 as follows: “Another possibility, which was not captured in this study, is cannabis users exposing pets to cannabis products while they were themselves intoxicated with cannabis.”

9. Line 389. Was repeat dose activated charcoal used in these cases or single dose activated charcoal.

From the data we collected, it is impossible to answer this question since the details of the treatments were not captured.

10. Line 398. ’Intra-lipids’ is a trade name (Intralipid®) so would be better to use intravenous lipid emulsion (IVLE).

We have revised the sentence on lines 431 – 433 as follows: “The administration of intravenous lipid emulsion was a treatment method reported by several participants in our study.” We also made similar changes in Fig 5b.

11. Figure 4. It is interesting you mention human faeces as a source of exposure. This paper has just been published.

Lauinger CA, Peacock R. Marijuana toxicosis in dogs in Melbourne, Australia, following suspected ingestion of human faeces: 15 cases (2011-2020). Aust Vet J. 2021 Nov 16. doi: 10.1111/avj.13134. Epub ahead of print.

We have added a sentence on lines 369 – 372 to this effect as follows: “It is interesting that we captured human feces ingestion as a source of cannabis toxicosis in pets in North America, since a recent study in Melbourne, Australia reported similar findings in dogs suggesting the need for pet owners to be cautious around accidental consumption, especially outdoors (43).”

Reviewer 2

1. This is a very interesting paper concerning a type of intoxication that seems to have some incidence in Canada and USA, with an increase after legalization of cannabis use in these countries. This can be relevant for other countries where this legalization occurs.

The information about clinical signs, products and route of exposure and diagnosis and treatment of the intoxication is also important and well reported and discuss.

The study have some limitation that were identified by the authors.

Only graphs from figure 2 and 3 are not so well done because x-axis is not explicit. Can the authors try to better explain these data?

Thank you for your comment. We have modified the titles of the graphs in Fig 2 and 3 and the titles of the x-axis in the graphs for both figures to improve clarity. Additionally, we have revised the text in the legends for both figures. Please see our responses for comments 2 and 5 from Reviewer 1.

---

## [Decision Letter · Decision Letter 1]

28 Mar 2022

Prevalence and Characteristics of Cannabis-induced Toxicoses in Pets: Results from a Survey of Veterinarians in North America

PONE-D-21-38974R1

Dear Dr. Khokhar,

We’re pleased to inform you that your manuscript has been judged scientifically suitable for publication and will be formally accepted for publication once it meets all outstanding technical requirements.

Kind regards,

Benito Soto-Blanco, DVM, MSc, PhD

Academic Editor

PLOS ONE

Reviewers' comments:

Reviewer's Responses to Questions

**Comments to the Author**

1. If the authors have adequately addressed your comments raised in a previous round of review and you feel that this manuscript is now acceptable for publication, you may indicate that here to bypass the “Comments to the Author” section, enter your conflict of interest statement in the “Confidential to Editor” section, and submit your "Accept" recommendation.

Reviewer #1: All comments have been addressed

Reviewer #2: All comments have been addressed

2. Is the manuscript technically sound, and do the data support the conclusions?

Reviewer #1: Yes

Reviewer #2: Yes

3. Has the statistical analysis been performed appropriately and rigorously? 

Reviewer #1: Yes

Reviewer #2: Yes

4. Have the authors made all data underlying the findings in their manuscript fully available?

Reviewer #1: Yes

Reviewer #2: Yes

5. Is the manuscript presented in an intelligible fashion and written in standard English?

Reviewer #1: Yes

Reviewer #2: Yes

6. Review Comments to the Author

Reviewer #1: Thank you for addressing the comments and making things clearer. The manuscript is much improved and I am happy to accept it for publication.

Reviewer #2: The information in the graphics was improved. And the information on the supplementary data are necessary and make the diference to the readers understand data presented on graphics.

7. PLOS authors have the option to publish the peer review history of their article (what does this mean?). If published, this will include your full peer review and any attached files.

Reviewer #1: No

Reviewer #2: No

---

## [Editor Report · Acceptance letter]

30 Mar 2022

PONE-D-21-38974R1 

Prevalence and Characteristics of Cannabis-induced Toxicoses in Pets: Results from a Survey of Veterinarians in North America 

Dear Dr. Khokhar:

I'm pleased to inform you that your manuscript has been deemed suitable for publication in PLOS ONE. Congratulations! Your manuscript is now with our production department. 

Kind regards, 

on behalf of

Dr. Benito Soto-Blanco 

Academic Editor

PLOS ONE